# Shallow Whole-Genome Sequencing of Cell-Free DNA (cfDNA) Detects Epithelial Ovarian Cancer and Predicts Patient Prognosis

**DOI:** 10.3390/cancers15020530

**Published:** 2023-01-15

**Authors:** Seong Eun Bak, Hanwool Kim, Jung Yoon Ho, Eun-Hae Cho, Junnam Lee, Sung Min Youn, Seong-Woo Park, Mi-Ryung Han, Soo Young Hur, Sung Jong Lee, Youn Jin Choi

**Affiliations:** 1Department of Obstetrics and Gynecology, Seoul St. Mary’s Hospital, College of Medicine, The Catholic University of Korea, 222, Banpo-daero, Seocho-gu, Seoul 06591, Republic of Korea; 2Cancer Research Institute, College of Medicine, The Catholic University of Korea, 222, Banpo-daero, Seocho-gu, Seoul 06591, Republic of Korea; 3Genome Research Center, GC Genome, 107, Ihyeon-ro 30beon-gil, Giheung-gu, Yongin 16924, Republic of Korea; 4Division of Life Sciences, College of Life Sciences and Bioengineering, Incheon National University, Incheon 22012, Republic of Korea

**Keywords:** cfDNA, epithelial ovarian cancer, shallow whole-genome sequencing, plasma, prognosis

## Abstract

**Simple Summary:**

Despite the progress in diagnostics and therapeutics, epithelial ovarian cancer (EOC) remains a fatal disease. Using shallow whole-genome sequencing (WGS), we identified copy number variations (CNVs). In addition, we quantified chromosomal instability using genome-wide instability and found that it could detect newly diagnosed EOC. In addition, the data showed *RAB25* amplification (alone or with CA125), and disease-free survival and overall survival. Our data demonstrated that cfDNA, detected by shallow WGS, represents a potential tool for diagnosing EOC and predicting its prognosis.

**Abstract:**

Despite the progress in diagnostics and therapeutics, epithelial ovarian cancer (EOC) remains a fatal disease. Using shallow whole-genome sequencing of plasma cell-free DNA (cfDNA), we investigated biomarkers that could detect EOC and predict survival. Plasma cfDNA from 40 EOC patients and 20 healthy subjects were analyzed by shallow whole-genome sequencing (WGS) to identify copy number variations (CNVs) and determine the Z-scores of genes. In addition, we also calculated the genome-wide scores (Gi scores) to quantify chromosomal instability. We found that the Gi scores could distinguish EOC patients from healthy subjects and identify various EOC histological subtypes (e.g., high-grade serous carcinoma). In addition, we characterized EOC CNVs and demonstrated a relationship between *RAB25* amplification (alone or with CA125), and disease-free survival and overall survival. This study identified *RAB25* amplification as a predictor of EOC patient survival. Moreover, we showed that Gi scores could detect EOC. These data demonstrated that cfDNA, detected by shallow WGS, represented a potential tool for diagnosing EOC and predicting its prognosis.

## 1. Introduction

Globally, 239,000 new epithelial ovarian cancer (EOC) cases are diagnosed yearly. In addition, 152,000 ovarian cancer (OC)-related deaths occur annually, making it the second leading cause of cancer-related death in females. Standard EOC treatment strategies include surgery and conventional chemotherapy [1,2]. However, EOC recurs in most patients (~70%), with a poor prognosis [3]. Thus, it is critical to understand the molecular pathways affecting prognosis. Indeed, markers that predict poor prognosis in EOC could help establish therapeutic strategies.

Homologous recombination repair (HRR) pathway deficiency leads to the chromosomal instability observed in EOC [4]. This HRR deficiency (HRD) results from germline or somatic mutations in *BRCA1/2* or other mechanisms. Increased platinum and PARPi (PARP inhibitor) responses occur due to pathogenic mutations in *BRCA1/2* and other genes, indicating its importance as a therapeutic target in solid tumors, including EOC [5]. This concept is illustrated by the central role of platinum agents in the management of EOC and the advent of PARPi [6]. The myriad myChoice^TM^ uses the combined HRD score (Genomic Instability Score) [7] in conjunction with *BRCA1/2* mutation and rearrangement analysis. This test has been included in several PARPi clinical trials and is FDA-approved as a companion diagnostic for niraparib and olaparib in relapsed EOC [5].

Most cancers are characterized by chromosomal instability (CIN), where there is a continual gain or loss of chromosomes or parts of chromosomes [8]. This process is often associated with poor prognosis. CIN in EOC and many other cancer types promotes tumor heterogeneity, clonal evolution, and chemotherapy resistance [9]. Copy number variations (CNVs), a CIN mechanism, are analyzed using high-throughput genome sequencing to detect carcinogenesis. CNVs consist of genomic segments of at least 50 bp that differ in copy number based on comparing two or more genomes [10]. An estimated 5–10% of DNA content can be spanned by amplifying or decreasing segments from 50 bp to longer than 1 kb, defined by copy number variations (CNVs) compared to a normal genome [11,12]. CNVs may cause changes in biological function (e.g., altered gene expression) and promote human disease [13,14].

EOC is characterized by aneuploidy genomes and a large burden of copy number amplifications and deletions [15]. In 2011, The Cancer Genome Atlas (TCGA) project identified *CCNE1, MYC*, and *MECOM* as common amplifications in EOC. Moreover, tumor suppressor genes (e.g., *PTEN*, *RB1*, and *NF1*) are frequently deleted in high-grade serous carcinoma (HGSC) [16], and CNVs predict overall survival (OS) and platinum-resistant relapse in HGSC, demonstrating their prognostic value [17]. For example, allelic *BRCA1* and *BRCA2* mutations combined with loss of the wild-type allele disrupt homologous recombination-mediated DNA damage repair, resulting in CNVs and loss of heterozygosity (LOH) [18]. Notably, genomic LOH correlated with the response to PARPi in a phase 2 clinical trial with epithelial EOC patients (ARIEL2). Taken together, CNVs may thus serve as potential clinical markers to predict cancer risk in various cancers [19].

Cell-free DNA (cfDNA) is released from tumor cells into the bloodstream and other bodily fluids [20]. This non-encapsulated circulating tumor DNA (ctDNA) is generated by multiple mechanisms, such as apoptosis, necrosis, and secretion from circulating tumor cells or extracellular vesicles [21]. Next-generation sequencing has shown that CNVs in cfDNA from plasma can explain the mechanisms underlying carcinogenesis and drug resistance in various cancers [22,23]. In addition, plasma cfDNA analysis is in the spotlight because it is a non-invasive tool to monitor cancer progression [21,24,25,26]. A prospective trial demonstrated that the recurrence rates for colorectal cancer were >10-fold higher in patients with detectable cfDNA postoperatively than those with undetectable cfDNA, identifying cfDNA as a predictive biomarker. However, there have been only limited studies on cfDNA in gynecologic cancer using whole genome sequencing (WGS) [27]. Therefore, we investigated the genetic characteristics of EOC and evaluated potential prognostic biomarkers in plasma cfDNA using WGS.

## 2. Materials and Methods

### 2.1. Samples

This study analyzed samples from 40 EOC patients and 20 healthy subjects (Table 1). Study participants provided written, informed consent. The study was approved by the Institutional Review Boards of Seoul St. Mary’s Hospital of the Catholic University of Korea College of Medicine (KC17TNSI0215) and Green Cross Laboratories (GCL-2017-1008-03). The healthy subjects were included for normalization (Appendix A). Blood (10 mL) was collected from each patient in EDTA tubes or Streck Cell-Free DNA BCT^®^ (Streck, La Vista, NE, USA) within one week before the surgery. All patients received systemic chemotherapy after the debulking surgery.

### 2.2. Preparation of cfDNA and NGS Data Preparation

cfDNA was extracted from plasma. For 30 EOC samples, extraction was performed with the QIAamp Circulating Nucleic Acid Kit (Qiagen, Valencia, CA, USA) using the manufacturer’s protocol. For ten EOC samples, cfDNA was extracted using the Chemagic cfDNA 2K Kit (PerkinElmer, MA, USA). The NGS library was quantified using the Qubit dsDNA HS assay kit (Thermo Fisher, MA, USA). Library fragment size was determined with D1000 screen tape (Agilent, CA, USA) using the Tapestation4200 (Agilent, CA, USA). Sequencing was performed with PE100 using the DNBSEQ-G400RS High-throughput Rapid Sequencing kit (MGI Tech Co., Shenzhen, China) and DNBSEQ-G400 sequencer (MGI Tech Co., Shenzhen, China).

### 2.3. Shallow WGS of cfDNA

Shallow WGS was performed at a mean depth of 0.27× (11 M reads/sample × [75 bp/read]/3 Gbp, the whole genome size). For sequences in which the Phred quality score was less than 33, adapter sequences were trimmed from the sequencing data using Atropos v1.1.28 before alignment [28]. Pre-processed reads were aligned to the hg19 reference genome using the mem algorithm of bwa v0.7.17. The alignments were sorted according to chromosome coordinates using samtools v1.0 (https://www.htslib.org/, accessed on 15 August 2014). PCR duplicates were eliminated with Picard MarkDuplicates v2.23.8 (https://broadinstitute.github.io/picard/, accessed on 15 October 2020), and alignments larger than the maximum insert size (8000), based on the samtools stats, were removed (Appendix A).

### 2.4. Data Processing for CNV Detection and Z-Score Grouping

Mosdepth v0.3.1 was used to check the read depth of OC-specific genes [29]. The LOESS algorithm was used to normalize GC bias [30]. The average and standard deviation of the gene body depth were calculated to determine the Z-scores using the 20 healthy subjects’ samples. Z-scores were calculated using the gene body depth of the patients. A Z-score exceeding 2 indicated amplification, and a score less than -2 signified a deletion.

To investigate whether OC-specific genes affected disease recurrence/progression, we performed Z-score grouping and two-step disease-free survival (DFS) analysis. The genes in the TCGA dataset were divided into AMP and DEL genes. AMP genes were defined as those with more amplified samples than deletion samples; DEL genes consisted of the genes with more deletion samples than amplified samples (Appendix A). Z-scores were grouped according to DEL and AMP information. For the AMP genes, Z-scores greater than 2 were grouped as 1, and Z-scores less than 2 were grouped as 0. For the DEL genes, Z-scores less than −2 were grouped as 1, and Z-scores greater than −2 were grouped as 0 (Appendix A).

### 2.5. Genome-Wide Instability Score

Genome-wide Z-scores were calculated to quantify chromosomal instability using the following equation [23]:Genome-wide instability score (Gi score) = ∑ *absolute*(*Z* − *score*)

### 2.6. Identification of OC-Specific Genes and CNV Validation Using the TCGA Dataset

The ovarian serous cystadenocarcinoma TCGA dataset (n = 489) was used to confirm the CNV pattern. The data were analyzed with the cBioPortal for Cancer Genomics (http://cbioportal.org, accessed on 10 November 2015) [31,32].

We evaluated the similarity of the copy number profiles of 33 genes obtained by shallow WGS using cfDNA and TCGA WGS using gDNA. Data for 95 ovarian tumor samples and 75 normal samples were downloaded from the TCGA database. The data/analyses presented in the current publication are based on the use of study data downloaded from the dbGaP website, under dbGaP accession number phs000178.v11.p8. The read depths of the target gene regions were determined using mosdepth v. 0.3.3. Normalization of the GC content was processed using the LOESS algorithm in R. Z-scores of the read depths were calculated using the mean and standard deviation from the normal samples. Genes with a Z-score that exceeded 2 were assumed to be amplified; those with a Z-score less than -2 were assumed to be deleted. Pearson correlation was used to investigate the correlation between the mean copy number from the cfDNA and that of the TCGA gDNA. All statistical analyses were performed with R-4.0.3 and visualized using the ggplot2 and ggpubr R packages (https://www.R-project.org, accessed on 24 April 2020 and https://CRAN.R-project.org/package=ggpubr, accessed on 27 June 2020) [30,33].

### 2.7. Statistical Analysis

DFS represents the time from treatment to relapse/progression. OS is the time to death, regardless of disease relapse/progression. Relapse is defined as a disease recurring more than six months after surgery; progression refers to a disease recurring within six months of surgery. DFS and OS from our data were determined by the Kaplan–Meier method. The genes with CNVs were analyzed using univariate analysis. For the multivariate analysis, the genes with CNVs and median CA125 level were used after classifying the patients into four subgroups: (1) genes with CNVs and CA125 > median CA125 (U/mL); (2) genes with CNVs and CA125 ≤ median CA125 (U/mL); (3) genes without CNVs and CA125 > median CA125 (U/mL); (4) genes without CNVs and CA125 ≤ median CA125 (U/mL). Inter-group comparisons were made using the log-rank test. Significant differences were defined as *p* < 0.05. The Wilcoxon rank sum test was used to analyze the statistical significance of the genome-wide instability scores (Gi scores) between groups.

## 3. Results

### 3.1. Clinical and Pathology Data of Subjects

This study included 40 epithelial EOC patients who received primary debulking surgery and adjuvant chemotherapy. The clinical-pathological characteristics of the patients are presented in Table 1. The median age at diagnosis was 54 years (range: 26–75). Nine patients had stage I disease, and two had stage II. The majority of the patients (73%, 29/40) were stage III/IV (III, n = 26; IV, n = 3). Most patients (57.5%, 23/40) were diagnosed with high-grade serous carcinoma (HGSC). The remaining patients were diagnosed with mucinous and clear cell carcinoma (n = 5 each), low-grade serous carcinoma (n = 3), and endometrioid carcinoma (n = 1).

During the follow-up period (median 50 months), twenty EOC patients (50%, 20/40) presented with disease recurrence/progression within a median time of 11 months following diagnosis (Table 1 and Appendix A). Six EOC patients (15%, 6/40) had *BRCA1/2* pathogenic variants, five of whom had disease recurrence/progression. The median follow-up time did not significantly differ between EOC patients with *BRCA1/2* pathogenic variants and wild-type *BRCA1/2* (53 months vs. 48 months).

### 3.2. Genome-Wide Z-Scores from Shallow WGS Detect Chromosomal Instability

Shallow whole-genome sequencing was performed to evaluate chromosomal instability in plasma cfDNA from EOC patients. For each patient sample, more than 79 M reads (175.56 ± 64.91 for all samples) were obtained; more than 110.6 M reads (200.52 ± 31.48 for all samples) were acquired for each normal sample. The coverage for each patient and normal subject was about 5.15× and 5.55×, respectively. To investigate the diagnostic performance of the shallow WGS CNV counts, the Z-scores for all CNVs were calculated for each sample. The heatmap shows the somatic amplifications in red (Z-score > 2) and deletions in blue (Z-score < −2) (Figure 1A). The CN profile showed the CN state across the genome in a predefined number of genes. Red segments represent CN gain in the genome-wide scatter plot using the Z-score, and green segments show CN loss (Figure 1B,C). Each dot represents a gene for which the copy number was inferred. In addition, we investigated the correlation of the 1× and 5× in silico analysis and found that the shallower WGS data resulted in similarly reliable outcomes (*R* = 0.8, *p* < 2.2 × 10^−16^) (Figure 1D). In this analysis, we downsampled the reads of the original data to 1x and calculated the correlation using the Z-scores of target genes.

### 3.3. Genome-Wide Instability by Shallow Whole-Genome Sequencing Characterizes EOC

We determined the genome-wide instability in cfDNA using the sum of absolute Z-scores (abs [Z-score]) (Appendix A). The genome-wide instability score (Gi score) for the EOC patients was significantly elevated compared to that of the healthy subjects (*p* = 0.0007, Wilcoxon test) (Figure 2A). In addition, the median Gi score of the advanced EOC patients (stage III/IV) was significantly higher than the median Gi score of the healthy subjects (*p* = 0.000579), whereas the median Gi score of the healthy subjects did not significantly differ from that of the early-stage EOC patients (stage I/II) (Figure 2B). There was a trend towards higher Gi scores for advanced-stage EOC patients compared to early-stage patients; however, the difference did not reach statistical significance. However, significant differences were observed between healthy subjects and HGSC (*p* = 0.02) (Figure 2C). The Gi score differences were not significant between the no recurrence/progression and recurrence/progression groups and between the wild-type *BRCA1/2* and *BRCA1/2* pathogenic variant groups (Appendix A).

To demonstrate the specificity of the observed CNVs identified in the cfDNA of EOC patients, we retrieved segmented CN data for ovarian tissue-derived gDNA in the TCGA database. The two datasets were in concordance based on the Z-scores, yielding a Spearman correlation coefficient of 0.44 (*p* = 0.0095) (Appendix A).

### 3.4. Copy-Number Variations in cfDNA predict EOC Patient Survival

We selected 33 EOC-specific genes from the TCGA dataset (n = 489, ovarian adenocarcinoma patients) (Table 2) [16]. Z-scores exceeding 2 indicated “amplification” (AMP); “deletion” (DEL) was defined by Z-scores less than −2 (Figure 1A–C). *MYC*, *MECOM*, *PRKCI*, *CCNE1*, and *EIF5A2* were the top five most commonly detected genes with CNVs in the TCGA dataset, and *AKT1, RPS6KA2, PIK3R1, EIF5A2*, and *PRKCI* were the top five from our data. Only *EIF5A2* and *PRKCI* were presented in the top five most commonly detected genes in both datasets.

We used Kaplan–Meier survival analysis to determine whether the results from the shallow WGS analysis of cfDNA were associated with clinical outcomes (i.e., DFS and OS). DFS analysis was performed using the 0/1 grouping information (0 = EOC patients with no relapse/progression, 1 = EOC patients with relapse/progression) (Appendix A). Patients who did not relapse/progress at the time of sample collection were treated as censored data. An event occurred when the patient gene grouping matched the TCGA CNV results. Of the 33 genes from the TCGA data, 10 genes (*RAB25*, *PIK3CA*, *MECOM*, *DLEC1*, *KIT*, *EGFR*, *CCNE1*, *PTEN*, *AKT1*, and *AKT2*) had p-values less than 0.05 (Appendix A). Four genes *(DLEC1*, *KIT*, *EGFR*, and *CCNE1*) occurred only once (event: n = 1) and were excluded (Figure 3).

OS analysis was performed using the following 0/1 grouping information: 0 = EOC patients who were alive; 1 = EOC patients who died. Three EOC patients were not available for follow-up. Therefore, only 37 of 40 EOC patients were analyzed (Appendix A). Three of the thirty-three genes from the TCGA dataset (*DPH1*, *ERBB2*, and *NOTCH3*) were excluded because none of the EOC patients harboring CNVs in these genes died during the follow-up period. We found four genes with CNVs (*RAB25*, *DAB2*, *TP53*, and *RPS6KA2*) present in more than three EOC patients, that demonstrated statistical significance in the OS analysis (*p* < 0.05) (Figure 4, Appendix A). In particular, we found that patients with amplified *RAB25* (*RAB25*_AMP, n = 3) had shorter DFS (*p* = 0.0014) and OS (*p* = 0.00011).

### 3.5. Integrated Value of cfDNA and CA125

The median preoperative CA125 concentration was 575.5 U/mL (range, 17.9–11,494.0) for all subjects and 954.9 U/mL (range, 26.0–11,494.0) and 105.0 U/mL (range, 17.9–21,760.0) for EOC patients with or without relapse/progression, respectively (Appendix A). We performed integrated analysis to evaluate the associations between the plasma cfDNA results and the median CA125 and clinical outcomes (DFS and OS). Kaplan–Meier analysis of the combinations of CA125 and nine genes with more than three events showed statistically significant effects on the DFS or OS (i.e., *RAB25*_AMP, *PIK3CA*_AMP, *MECOM*_AMP, *PTEN*_DEL, *AKT1*_AMP, *AKT2*_AMP, *DAB2*_AMP, *TP53*_DEL, and *RPS6KA2*_DEL) (Appendix A). The four subgroups analyzed were (1) genes with CNVs and CA125 > 575.5 U/mL; (2) genes with CNVs and CA125 ≤ 575.5 U/mL; (3) genes without CNVs and CA125 > 575.5 U/mL; (4) genes without CNVs and CA125 (U/mL) ≤ 575.5 U/mL.

For the DFS, the six genes with CNVs (*RAB25, PIK3CA, MECOM, PTEN, AKT2*, and *RPS6KA2*) and CA125 (subgroup 1) had the shortest DFS rate of the four subgroups (*p* < 0.05) (Figure 5A and Appendix A). In contrast, all six genes without CNVs and CA125 (subgroup 4) had the longest DFS rate (*p* < 0.05). For subgroup 2, only five genes contained CNVs (*PIK3CA, MECOM, PTEN, AKT2*, and *RPS6KA2*). These genes had the second shortest DFS rate. Subgroup 3 had the third shortest DFS rate. We performed OS analysis using the six genes from the DFS analysis. The results showed that only “*RAB25_*AMP and CA125 > 575.5 U/mL” had the shortest OS survival rate reaching statistical significance (*p* = 0.00056) (Figure 5B).

## 4. Discussion

In this study, we investigated whether chromosomal instability in cfDNA, detected by shallow WGS, is a potential marker for EOC and a prognostic indicator. We found that plasma cfDNA could detect genome-wide instability in EOC. Our data also showed that CNVs could provide comprehensive genetic data for EOC patients. Moreover, *RAB25* amplification predicted EOC patient survival. Our study is the first to reveal that *RAB25* amplification in cfDNA, identified by shallow-WGS, can predict EOC patient prognosis.

Using shallow WGS, we demonstrated that detecting chromosomal instability in cfDNA from EOC patients is feasible. We investigated the correlation of 1× and 5× in silico analysis and found that the shallower WGS data produced similarly reliable outcomes (Figure 1D). Next, we quantified genome-wide instability by developing a “genome-wide instability score” or Gi score using the sum of (abs[Z-score]). EOC patients had higher Gi scores than healthy subjects. In addition, we observed a trend towards higher Gi scores in advanced-stage EOC patients compared to early-stage patients. Our results are consistent with recent studies showing that cfDNA detected by shallow WGS signifies chromosomal instability in EOC patients [34,35,36].

Our data showed that CNV data from cfDNA, obtained from shallow WGS, provides comprehensive genetic data for EOC, and may be a highly specific biomarker reflecting chromosomal instability. In addition, we determined that CNVs from shallow WGS could predict EOC prognosis. Previous studies showed that WGS makes it possible to detect cancer-specific CNVs in cfDNA [34,35,37,38]; however, studies predicting survival outcomes by this method are limited. Vanderstchele et al., were the first to evaluate the potential of cfDNA for diagnosing EOC. Braicu et al. showed that cfDNA quantification using shallow WGS based on CNVs could be used for EOC diagnosis and treatment monitoring [34,36]. Some studies demonstrated an association between chromosomal instability in cfDNA detected by shallow WGS and EOC prognosis. However, one study could only demonstrate the clinical usefulness of shallow WGS in prognosis prediction when it was combined with whole exome sequencing (WES). Because of the high cost of WES, combining shallow WGS with WES for determining patient prognosis would negate the financial benefits offered by shallow WGS [39]. Another study suggested that shallow WGS may be a clinically feasible method for predicting survival; however, it did not identify an association between chromosomal instability and EOC prognosis on a genetic level [35].

In our study, *RAB25* amplification was associated with poor DFS and OS. In addition, we found that the integrated value for *RAB25* amplification and the known tumor marker CA125 was associated with EOC patient prognosis. *RAB25* was also amplified in the EOC TCGA dataset [16] and is considered an oncogene in various malignancies, including EOC [40,41]. We have limitations that the ages of the healthy subjects and EOC patients were not matched, and the family histories of the healthy subjects were not provided. However, previous studies presented that ~0.25% of the general population harbored germline *BRCA1*/*2* mutations; therefore, it is suggested that the healthy subjects should be considered without germline *BRCA1*/*2* mutations [42,43,44,45]. In addition, this study had a limited number of EOC patients with amplified *RAB25* (7%; n = 7/40) and the overall sample size was small. However, the median follow-up period for the EOC patients in this study was longer (50 months) than previous studies, and this may demonstrate the reliability of the study [35].

## 5. Conclusions

We demonstrated, for the first time, that *RAB25* amplification is predictive of EOC patient survival. It also showed that the genome-wide instability score could detect EOC. Shallow WGS is an updated tool that can use cfDNA, which can be obtained non-invasively. This approach has a cost-benefit, over WGS and WES. However, further validation of the approach with large cohort studies is required. Collectively, our data showed that cfDNA detected using shallow WGS may be a clinically applicable tool for diagnosing EOC and predicting patient prognosis.

## Figures and Tables

**Figure 1 cancers-15-00530-f001:**
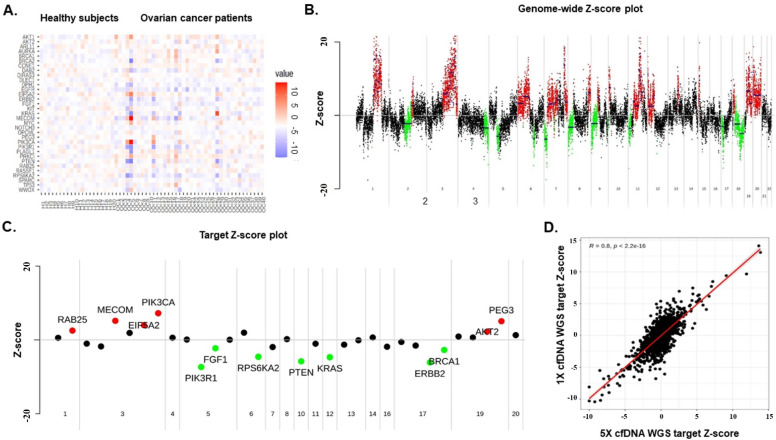
(**A**) The heatmap of the shallow WGS for 40 EOC patients and 20 healthy subjects. Amplified genomic regions are shown in red, and deleted genomic regions are presented in blue. These regions are based on window Z-score calculations. (**B**) Genome-wide scatter plot using the Z-scores (OC11). Scatter plot using the Z-scores with the bins for the segmented genome set to 100,000 bases. Red are the segments with copy gain, and green are those with copy loss. (**C**) Target gene scatter plot using the Z-scores (OC11). Scatter plot using the Z-scores of the target genes for each sample (red circle, Z-score ≥ 2; green circle, Z-score < 2; black circle, other). (**D**) Correlation plot using an in-silico method. Downsampling of the reads of the original data to 1X. The correlation was calculated using the Z-scores of the target genes.

**Figure 2 cancers-15-00530-f002:**
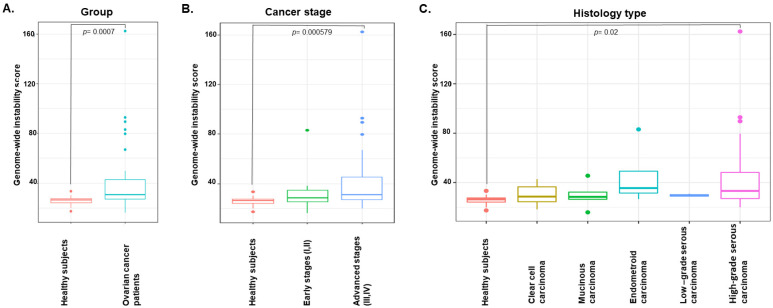
Distribution of genome-wide instability scores according to participant type (**A**) (OC patients vs. healthy subjects), (**B**) cancer stage (stages I/II [early] vs. stages III/IV [advanced]), and (**C**) histology type.

**Figure 3 cancers-15-00530-f003:**
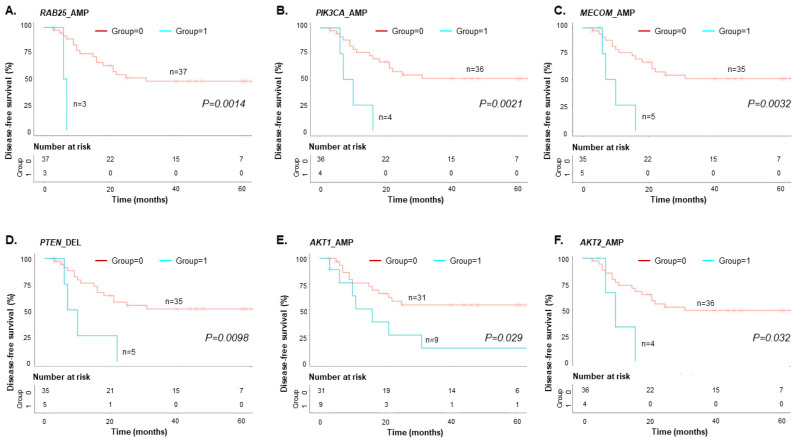
Kaplan–Meier analysis for EOC patient DFS (n = 40). (**A**) *RAB25* amplification (*RAB25*_AMP), (**B**) *PIK3CA_AMP*, (**C**) *MECOM*_AMP, (**D**) *PTEN* deletion (*PTEN*_DEL), (**E**) *AKT1*_AMP, and (**F**) *AKT2*_AMP had a significantly worse DFS (*p* < 0.05, log-rank test).

**Figure 4 cancers-15-00530-f004:**
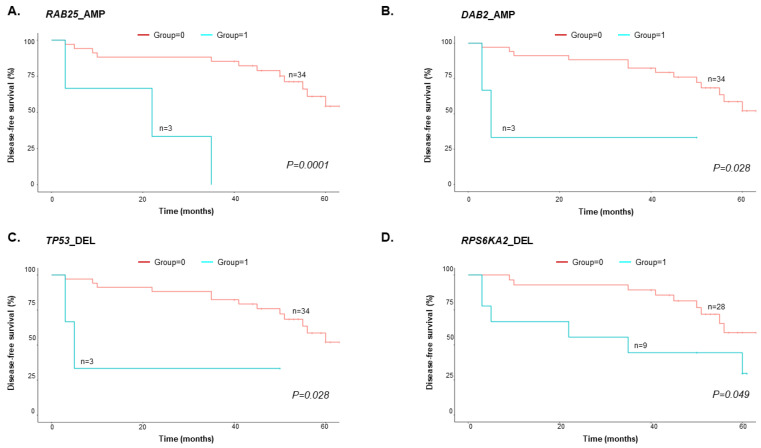
Kaplan–Meier analysis of EOC patient OS (n = 37). (**A**) *RAB25*_AMP, (**B**) *DAB2_*AMP, (**C**) *TP53*_DEL, and (**D**) *RPS6KA2_*DEL had significantly worse OS (*p* < 0.05, log-rank test).

**Figure 5 cancers-15-00530-f005:**
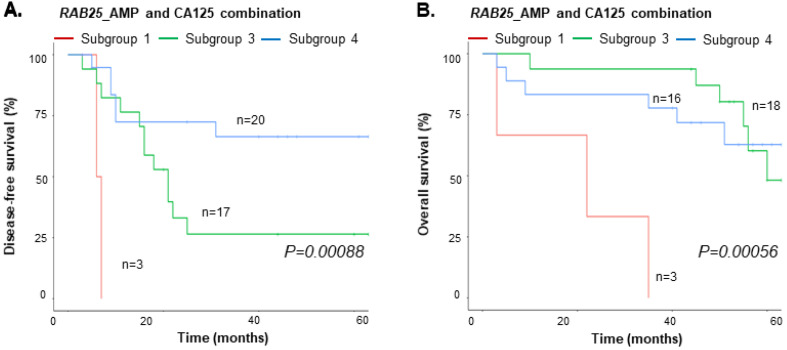
cfDNA and CA125 combination analysis. (**A**) DFS and (**B**) OS of EOC patients with *RAB25* amplification and CA125.

**Table 1 cancers-15-00530-t001:** Clinico-pathological characteristics of the ovarian cancer patients.

	Ovarian Cancer Patients (n = 40)
Age (median) [quartile 1; quartile 3]	54 yr [47.5; 61.5]
FIGO stage	
I	9
II	2
III	26
IV	3
BRCA mutation	
Yes	6
No	34
Pathologic types	
High-grade serous carcinoma	23
Low-grade serous carcinoma	3
Mucinous carcinoma	5
Endometrioid carcinoma	4
Clear cell carcinoma	5
Recurrence/ progression	
Yes	20
No	20
Disease-free time (median) [quartile 1; quartile 3]	11 mo [7; 19.5]
Follow- up time (median) [quartile 1; quartile 3]	50 mo [37.5; 58]
CA125 (U/mL) (median) [quartile 1; quartile 3]	575.5 [91.5; 1175.8]

**Table 2 cancers-15-00530-t002:** Copy number variation (CNV) pattern of ovarian cancer-specific oncogenes and tumor suppressor genes from TCGA data.

Chr	Start (hg19)	End (hg19)	Gene	Strand	CNV	Occuring Events_Frequency	TCGA_Frequency	Oncogenic Information
chr1	156,030,965	156,040,295	*RAB25*	+	AMP	7.50%	7.20%	Oncogene
chr3	168,801,286	169,381,563	*MECOM*	−	AMP	12.50%	24.70%	Oncogene
chr3	169,940,219	170,023,770	*PRKCI*	+	AMP	15.00%	22.10%	Oncogene
chr3	170,606,203	170,626,426	*EIF5A2*	−	AMP	22.50%	20.70%	Oncogene
chr3	178,866,310	178,952,497	*PIK3CA*	+	AMP	10.00%	18.00%	Oncogene
chr4	55,524,094	55,606,881	*KIT*	+	AMP	2.50%	1.20%	Oncogene
chr5	67,511,583	67,597,649	*PIK3R1*	+	DEL	20.00%	2.00%	Oncogene
chr5	141,971,742	142,077,635	*FGF1*	−	AMP	5.00%	1.00%	Oncogene
chr7	55,086,724	55,275,031	*EGFR*	+	AMP	2.50%	0.40%	Oncogene
chr8	128,748,314	128,753,680	*MYC*	+	AMP	7.50%	31.50%	Oncogene
chr12	25,358,179	25,403,854	*KRAS*	−	AMP	2.50%	9.80%	Oncogene
chr14	105,235,686	105,262,080	*AKT1*	−	AMP	22.50%	2.90%	Oncogene
chr17	37,844,392	37,884,915	*ERBB2*	+	AMP	0.00%	2.20%	Oncogene
chr19	15,270,443	15,311,792	*NOTCH3*	−	AMP	0.00%	11.50%	Oncogene
chr19	30,302,900	30,315,215	*CCNE1*	+	AMP	2.50%	21.70%	Oncogene
chr19	40,736,223	40,791,302	*AKT2*	−	AMP	10.00%	7.00%	Oncogene
chr20	54,944,444	54,967,351	*AURKA*	−	AMP	10.00%	3.90%	Oncogene
chr1	68,511,644	68,516,460	*DIRAS3*	−	AMP	2.50%	1.00%	Tumor suppressor gene
chr3	38,080,695	38,164,228	*DLEC1*	+	AMP	2.50%	0.60%	Tumor suppressor gene
chr3	50,367,216	50,378,367	*RASSF1*	−	AMP	7.50%	1.00%	Tumor suppressor gene
chr5	39,371,775	39,425,335	*DAB2*	−	AMP	10.00%	3.10%	Tumor suppressor gene
chr5	151,040,656	151,066,615	*SPARC*	−	AMP	2.50%	1.20%	Tumor suppressor gene
chr6	144,261,436	144,385,735	*PLAGL1*	−	DEL	12.50%	0.60%	Tumor suppressor gene
chr6	166,822,853	167,275,771	*RPS6KA2*	−	DEL	22.50%	1.40%	Tumor suppressor gene
chr10	89,623,194	89,728,532	*PTEN*	+	DEL	12.50%	6.10%	Tumor suppressor gene
chr11	132,284,874	133,402,403	*OPCML*	−	AMP	7.50%	2.50%	Tumor suppressor gene
chr13	32,889,616	32,973,809	*BRCA2*	+	DEL	10.00%	0.80%	Tumor suppressor gene
chr13	50,202,434	50,208,008	*ARL11*	+	DEL	2.50%	2.20%	Tumor suppressor gene
chr16	78,133,326	79,246,564	*WWOX*	+	DEL	10.00%	5.70%	Tumor suppressor gene
chr17	1,933,430	1,946,725	*DPH1*	+	AMP	0.00%	1.00%	Tumor suppressor gene
chr17	7,571,719	7,590,868	*TP53*	−	DEL	10.00%	0.60%	Tumor suppressor gene
chr17	41,196,311	41,277,500	*BRCA1*	−	DEL	7.50%	0.60%	Tumor suppressor gene
chr19	57,321,444	57,352,094	*PEG3*	−	AMP	5.00%	1.80%	Tumor suppressor gene

## Data Availability

The data presented in this study are available on request from the corresponding author (Y.J.C.). The data are not publicly available due to ethical restrictions.

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
