# Peer review of "Shallow Whole-Genome Sequencing of Cell-Free DNA (cfDNA) Detects Epithelial Ovarian Cancer and Predicts Patient Prognosis"

_cancers, 2023, doi:10.3390/cancers15020530_

Round 1

Reviewer 1 Report

Shallow whole-genome sequencing (WGS) cfDNA detects ovarian cancer and predicts prognosis

This nicely done study is from a reputable Korean team with interesting results and confirmation that this is a technology with potentially important clinical merit.

The authors should not claim to be the first to link cfDNA ovarian cancer and prognosis (Pubmed 68 hits) and the discussion should build on such papers (Which should also be cited) as Thusgaard CF, Korsholm M, Koldby KM, et al. Epithelial ovarian cancer and the use of circulating tumor DNA: A systematic review. Gynecol Oncol. 2021;161(3):884-895, Paracchini L, Beltrame L, Grassi T, et al. Genome-wide Copy-number Alterations in Circulating Tumor DNA as a Novel Biomarker for Patients with High-grade Serous Ovarian Cancer. Clin Cancer Res. 2021;27(9):2549-2559, and Steffensen KD, Madsen CV, Andersen RF, et al. Prognostic importance of cell-free DNA in chemotherapy resistant ovarian cancer treated with bevacizumab. Eur J Cancer. 2014;50(15):2611-8.

Why do the authors think the BRCA mutated tumors did not have a better prognosis?

Minor issues:

Pg 1 ln 32 truncated sentence

Pg 1 ln 40 missing space cancer(OC)

Pg 2 ln 52 myChoice should have a superscripted TM

Author Response

Reviewer #1>

Comment 1 : The authors should not claim to be the first to link cfDNA ovarian cancer and prognosis (Pubmed 68 hits) and the discussion should build on such papers (Which should also be cited) as Thusgaard CF, Korsholm M, Koldby KM, et al. Epithelial ovarian cancer and the use of circulating tumor DNA: A systematic review. Gynecol Oncol. 2021;161(3):884-895, Paracchini L, Beltrame L, Grassi T, et al. Genome-wide Copy-number Alterations in Circulating Tumor DNA as a Novel Biomarker for Patients with High-grade Serous Ovarian Cancer. Clin Cancer Res. 2021;27(9):2549-2559, and Steffensen KD, Madsen CV, Andersen RF, et al. Prognostic importance of cell-free DNA in chemotherapy resistant ovarian cancer treated with bevacizumab. Eur J Cancer. 2014;50(15):2611-8.

Response 1: Thank you for the comment. We changed the sentence “This study is the first to identify RAB25 amplification~~” to “This study identified RAB25 amplification ~” (page 1, line 30). In addition, we also changed the sentence “Our study is the first to reveal that CNVs can~” to “. Our study is the first to reveal that RAB25 amplification in cfDNA that was identified by shallow-WGS can~” (page 11, line 306. We have cited the paper “Genome-wide Copy-number Alterations in Circulating Tumor DNA as a Novel Biomarker for Patients with High-grade Serous Ovarian Cancer. Clin Cancer Res. 2021;27(9):2549-2559” as reference [ 35]. In addition, we have added the references “Thusgaard CF, Korsholm M, Koldby KM, et al. Epithelial ovarian cancer and the use of circulating tumor DNA: A systematic review. Gynecol Oncol. 2021;161(3):884-895” [25] and “Steffensen KD, Madsen CV, Andersen RF, et al. Prognostic importance of cell-free DNA in chemotherapy resistant ovarian cancer treated with bevacizumab. Eur J Cancer. 2014;50(15):2611-8.” [26].

Comment 2: Why do the authors think the BRCA mutated tumors did not have a better prognosis?

Response 2: There has been a number of studies that attempted to identify the association between the survival and BRCA1/2 mutations in ovarian cancer. However, the results are controversial. Some studies presented that the OC patients with BRCA1/2 mutations have better prognosis [JAMA. 2012 Jan 25;307(4):382-90] where as others presented that the BRCA1/2 mutations may affect a short-term survival but not a long-term survival [J Natl Cancer Inst. 2013 Jan 16;105(2):141-8]. Moreover, it is suggested that may be small number of the participants may be the reason why the BRCA1/2 mutated tumors did not have a better prognosis. In addition, the OC patients with BRCA1/2 mutations included in this study did not take PARP inhibitors which are known to prolong the long-term survival [JCO.2022.40.16_suppl.e18812].

Comment 3 Minor issues:

Response 3: Thank you for the comment and we edited as suggested

1) Pg 1 ln 32 truncated sentence: We changed “WGS represents a” to “WGS represented a potential tool for diagnosing EOC and predicting its prognosis.”.

2) Pg 1 ln 40 missing space cancer(OC): We changed “cancer(OC)” to “cancer (OC)”.

3) Pg 2 ln 52 myChoice should have a superscripted TM: we changed “myChoice” to “myChoiceTM” for clarification.

Reviewer 2 Report

My dears,

Please find my comments in the attached pdf.

The idea behind the work is good but there was a lack of general care and attention towards various aspects of the paper rendering it hard to follow and thus giving me many doubts while reading it. Patients grouping is unclear in the present form, as well as the subsequent analyses, results are only partially validated on TCGA without further commenting on the differences or even lack of complete analyses.

You haven't proven that your healthy group is actually healthy. Moreover they are significantly younger thus can bear BRCAmutations and develop cancer later in life. You could have al least chosen age-matched women...

Other: Make sure fonts are consistent. References are missing or improperly cited. Sentences are cropped. Sometimes the sentences are just thrown there without a general nice and enjoyable flow of ideas.

I look forward to you revised manuscript.

Author Response

January 1st, 2023

Editors, Cancers

Dear Sir/Madam,

On December 14th, a Decision Letter for the manuscript, cancers-2103211, entitled “Shallow whole-genome sequencing of cell-free DNA (cfDNA) identifies epithelial ovarian cancer and predicts the prognosis” was received. The manuscript has been reviewed and revised in accordance with the reviewers’ comments. The corrected sections in the revised manuscript are marked in yellow.

Reviewer #2>

Comment 1 : we [Pg 1 In 15]

Response 1: As suggested in the paper, we changed “Using shallow whole-genome sequencing (WGS) identified copy number variations (CNVs)~” to “Using shallow whole-genome sequencing (WGS), we identified copy number variations (CNVs)~”.

Comment 2: I think such details are too complex for a simple summary. Consider keeping it simple while enhancing the main findings of your work. [Pg 1 In 16]

Response 2: Thank you for the comment. As suggested we changed “~ copy number variations (CNVs) and determined the Z-scores of genes. We found that Genome-wide instability score (Gi scores) (=Σ ????????(?−?????)) ~.” to “ ~copy number variations (CNVs). In addition, we quantified chromosomal instability using Genome-wide instability and found that it could~ ”

Comment 3 : in recurrence or newly diagnosed patients? [ Pg 1 In 31]

Response 3: It seems that the sentence was not clear therefore, we changed the sentence from “ ~ could detect EOC.” to “ ~ could detect newly diagnosed EOC”.

Comment 4 : this should be deleted [Pg 1 In 32,33]

Response 4 : Thank you for the comment. We deleted the words “Keywords: keyword 1; keyword 2; keyword 3 (List three to ten pertinent keywords specific to the article yet reasonably common within the subject discipline.)”

Comment 5 : this sentence has been cropped. [Pg 1 In 32]

Response 5: We changed “WGS represents a” to “WGS represented a potential tool for diagnosing EOC and predicting its prognosis.”.

Comment 6 : this part would benefit from some rephrasing. most sentences are short and look like they were cropped with an axe. Some elegance would improve this :) for example you can use some more connectors like: moreover, first, one one hand, etc etc [Pg 2 In 67~76]

Response 6 : We appreciate this advice. We have changed the text from “EOC is characterized by aneuploidy genomes and a large burden of copy number amplifications and deletions [15]. In 2011, The Cancer Genome Atlas (TCGA) project identified CCNE1, MYC, and MECOM as common amplifications in EOC. Tumor suppressor genes (e.g., PTEN, RB1, and NF1) are frequently deleted in high-grade serous carcinoma (HGSC) [16]. Indeed, CNVs predicted overall survival (OS) and platinum-resistant relapse in HGSC, demonstrating their prognostic value [17]. For example, allelic BRCA1 and BRCA2 mutations combined with the loss of the wild-type allele disrupt homologous recombination-mediated DNA damage repair, resulting in CNVs and loss of heterozygosity (LOH) [18]. Notably, genomic LOH correlated with the response to PARPi in a phase 2 clinical trial with epithelial EOC patients (ARIEL2). CNVs may serve as potential clinical markers to predict cancer risk in various cancers [19]. HER2 amplification is predictive in advanced gastric cancer patients [20].“ to “EOC is characterized by aneuploidy genomes and a large burden of copy number amplifications and deletions [15]. In 2011, The Cancer Genome Atlas (TCGA) project identified CCNE1, MYC, and MECOM as common amplifications in EOC. Moreover, tumor suppressor genes (e.g., PTEN, RB1, and NF1) are frequently deleted in high-grade serous carcinoma (HGSC) [16], and CNVs predict overall survival (OS) and platinum-resistant relapse in HGSC, demonstrating their prognostic value [17]. For example, allelic BRCA1 and BRCA2 mutations combined with loss of the wild-type allele disrupt homologous recombination-mediated DNA damage repair, resulting in CNVs and loss of heterozygosity (LOH) [18]. Notably, genomic LOH correlated with the response to PARPi in a phase 2 clinical trial with epithelial EOC patients (ARIEL2). Taken together, CNVs may thus serve as potential clinical markers to predict cancer risk in various cancers [19].”

In addition, we have gone through English editing process and the below is the certificate from the company.

Comment 7: you can make this one sentence [Pg 2 In 68~70]]

Response 7 : Thank you for the suggestion. We have combined the two sentences.

Comment 8: the way this info is parked here it's completely out of place. [Pg2 In 77]

Response 8: Thank you for the comment. We erased the sentence “HER2 amplification is predictive in advanced gastric cancer patients [20].”

Comment 9: you need to add some info on the patients. selection criteria, clinical data extraction, follow up period, etc [Pg 2 In 95]

Response 9: We have the above mentioned data in “Results 3.1” (page 4). We wrote “This study included 40 epithelial EOC patients who received primary debulking surgery and adjuvant chemotherapy. The clinical-pathological characteristics of the patients are presented in Table 1. The median age at diagnosis was 54 years (range: 26–75). Nine patients had stage I disease, and two were at stage II. The majority of the patients (73%, 29/40) were stage III/IV (III, n = 26; IV, n = 3). Most patients (57.5%, 23/40) were diagnosed with high-grade serous carcinoma (HGSC). The remaining patients were diagnosed with mucinous and clear cell carcinoma (n = 5 each), low-grade serous carcinoma (n = 3), and endometrioid carcinoma (n = 1).

During the follow-up period (median 50 months), twenty EOC patients (50%, 20/40) presented with disease recurrence/progression within a median time of 11 months following diagnosis (Table 1 and Table S1). Six EOC patients (15%, 6/40) had BRCA1/2 pathogenic variants, five of which had disease recurrence/progression. The median follow-up time did not significantly differ between EOC patients with BRCA1/2 pathogenic variants and wild-type BRCA1/2 (53 months vs. 48 months).”

However for clarification, we changed “This study analyzed samples from 40 EOC patients and 20 healthy subjects.” to “This study analyzed samples from 40 EOC patients and 20 healthy subjects (Table 1)” (page 2, line 95).

Comment 10: so they were all upfront patients? what was the residual disease? some other surgical details would be also welcomed in table 1. also some details on the healthy subject. How did you define them as healthy? normal ultrasound, normal CA125, no BRCA? their age? I saw the supplementary files and the CA and BRCA is not present. This data is important to define your control group since they are not even age matched! [Pg 3 In 101]

Response 10: Healthy subjects were volunteers and selected through a brief health survey about cancer diagnosis, organ transplant, and blood transfusion within the last 5 years. We did not perform BRCA1/2 or CA125 tests. The average age of healthy subjects was 35.5 years, and the average age of EOC samples was 55.6 years. Although the ages were not matched between the control and EOC group, we have the survey data that affirms they are healthy subjects. In addition, large published datasets (e.g. GTEx [Nat. Genet. 45, 580–585 (2013)]) that a number of investigators use for studies lack publicly available data on the donor’s age and thus age-matched studies are impossible. From this information, we suggest that age-matching may not be compulsory in this study.

Comment 11: I would rename this something like CNV validation on TCGA-OV and leave it towards the end of the methods so to respect a bit the flow of the ideas in the study [Pg 3 In 103]

Response 11: As suggested we erased the “2.2 Identification of OC specific genes” and merged them into “2.6 Identification of OC-specific genes and CNV validation using the TCGA dataset”. Therefore the below sentences were included in the section 2.6 (page 4). “The ovarian serous cystadenocarcinoma TCGA dataset (n = 489) was used to confirm the CNV pattern. The data were analyzed with the cBioPortal for Cancer Genomics (http://cbioportal.org) [27, 28].” In addition, the subtitle numbers were changed (e.g. 2.3->2.2, 2.4--> 2.3…).

Comment 12 : why this? this can be a source of bias why mix up protocols? do you have some samples done with both kits to show that there is no difference is the results? [Pg 3 In 108~110]

Response 12: There is no result of direct comparison by analyzing the same sample with QIAmp and Chemagic 2 kits. However, there is a comparison between the Tiangen micro DNA kit (Tiangen Biotech, Beijing, China) which use the same principle as the QIAamp kit, the column method and the chemagic kit. There was no difference as a result of analyzing the same sample with both kits. When comparing the Z-score of cancer patients (liver cancer), it is analyzed similarly with R2=0.98 (below figure). Also, when calculating genome-wide instability score, it is similarly calculated as 136.9 in Tiagne kit and 135.8 in Cheamgic kit. There will be no difference in genome-wide instability analysis results between the column-type QIAamp and the bead-type Chemagic kit.

Comment 13: so you got access to the controlled data I assume? I believe you have to cite the dbGaP approval here. [Pg 4 In 149]

Response 13 : We added the below sentence in page 4. “The data/analyses presented in the current publication are based on the use of study data downloaded from the dbGaP web site, under dbGaP accession number phs000178.v11.p8.”

Comment 14 : add: continuous variables are reported as mean st dev, or median quartiles and categorical variables as absolute numbers and frequencies. (or something similar. Ofc you have to actually do this in the tables as you usually have the mean only.) [Pg 4 In 159]

Response As suggested, we added median quartiles for the variables “age, disease-free time, follow-up time and CA125 (U/ml) in Table 1. In addition, we re-checked the median follow-up time and changed it from 54 mo to 50 mo.

Comment 15: also state where the TCGA clinical data was taken from? [Pg 4 In 163]

Response 15 : We did not obtain TCGA clinical data. Disease-free survival and overall survival data were only from our own dataset. For clarification, we changed “DFS and OS~” to “DFS and OS from our data”.

Comment 16 : what stages are the histologies? It's not a good idea to make such a study on such a heterogeneous cohort. [Pg 4 In 191]

Response 16 : The below table presents the histologies and stages. We attempted to include homogenous cohort but since it is a real-world data the clinico-pathologic data were somewhat heterogeneous. However, since we have the raw-data in the supplementary TableS1, we did not insert the below table in the main manuscript.

Ovarian cancer patients (n=40)

Age (median)

54 yrs

FIGO stage

 I

9

 II

2

 III

26

 IV

3

BRCA mutation

 Yes

6

 No

34

Pathologic types

 High-grade serous carcinoma

23

stage I

0

stage II

0

stage III

21

stage IV

2

 Low-grade serous carcinoma

3

stage I

0

stage II

0

stage III

3

stage IV

0

 Mucinous carcinoma

5

stage I

3

stage II

1

stage III

0

stage IV

1

 Endometrioid carcinoma

4

stage I

3

stage II

1

stage III

0

stage IV

0

 Clear cell carcinoma

5

stage I

3

stage II

0

stage III

2

stage IV

0

Recurrence/ progression

 Yes

20

 No

20

Disease-free time (median)

11 mo

Follow- up time (median)

50 mo

CA125 (U/ml) (median)

575.5

Comment 17: so this is only one sample? I think you should do this control on more samples... and show it in supplementary [Pg 6 In 212]

Response 17:We attach the JPG files that present Genome-wide scatter plot using the Z-scores (Figure 1B) and Target gene scatter plot 211 using the Z-scores (Figure 1C) of all the samples (File name: Reviewer2_comment17).

Comment 18: how do you justify this difference? MYC at least should have been in this list since everyone finds at least this one... [Pg 7 In 242~244]

Response 18 : We analyzed the TCGA and our data and the differences were written in Table 2. Please look at the column “Occuring events frequency” and “TCGA frequency”.

Comment 19 : hg19 positions? mention ref genome [Pg 7 Table2 ]

Response 19 : Yes, it was hg 19. For clarification, we added (hg19) in column “Start” and “End” of Table 2.

Comment 20: move this in the legend [Pg 8 In 249]

Response 20: As suggested, we moved “Chr, chromosome;AMP,amplification;DEL,deletion.”

Comment 21 : Since you are doing multiple comparisons of all genes I believe a adjusted p value would be required. Please comment on the lack of adjustment so to make clear whether your choice is good or needs to be reconsidered [Pg 8 In 257]

Response 21: We analyzed the statistics of the Figure 2. For Figure 2A, the adjusted p-value was the same as the previous one, but the statistics for Figures 2B and 2C changed as below. Therefore, we changed the Figure 2 and the related numbers in page 5 (Results, 3.3 Genome-wide instability by shallow whole-genome sequencing characterizes EOC).

Comment 22: please add the risk tables and increase the visibility of the censored points from the curves [Pg 9 In 262]

Response 22: We changed the Figures 3 and 4 that include the risk tables as suggested (below and the main manuscript).

Figure 3.

Figure 4.

Comment 23 : are these CNV occurring together? Seems like some of them at least are the same patients [Pg 10 In 275]

Response 23 : Yes, they are occurring together. The data in detail is written in Table S5.

Comment 24 : one event as a threshold of inclusion is really really low. I would keep only those with at least 3/group... [ Pg 10 In 283]

Response 24 : Thank you for the suggestion. As suggested we analyzed the genes with more than three events and described in the manuscript (pages 9 and 10).

Comment 25 : why is the title of these figures RAB25_AMP? there are other groups plotter here too... [Pg 11 In 299]

Response 25 : For clarification, we changed the “RAB25_AMP” in the figure to “RAB25_AMP and CA125 combination”.

Comment 26 : Reference number [Pg 11 In 322]

Response 26 : We checked the reference list and added the number at the end of sentence (references 25 and 26 are the new ones and they are marked in yellow, in “Reference” part.).

Comment 27: The idea behind the work is good but there was a lack of general care and attention towards various aspects of the paper rendering it hard to follow and thus giving me many doubts while reading it. Patients grouping is unclear in the present form, as well as the subsequent analyses, results are only partially validated on TCGA without further commenting on the differences or even lack of complete analyses (page 8, 257).

Response 27: To confirm the CNV pattern, we first downloaded the CNV frequency of 33 EOC-specific genes from the cBioportal (n = 489 ovarian adenocarcinoma patients). These were pre-computed results using TCGA data. Then, we decided to evaluate the similarity of the copy number profiles of 33 genes obtained by shallow WGS (cfDNA) and TCGA (gDNA). Due to the TCGA raw data availability, we only used 95 ovarian tumor samples and 75 normal samples.

Round 2

Reviewer 2 Report

Dear authors,

Thank you for taking the time to go through my comments and consider my suggestions.

Last requests:

genes should be written in italic

Regarding the age matching:

GTEx samples have the age. Your response is not satisfactory and I'm sure you know that too... Choosing a matched control is always recommended by any statistician to prevent potential analysis biases. Sometimes even propensity score matching is employed to ensure a good selection of the subjects. Age matching is the easiest thing to do. You do agree that any of those healthy subjects can have a BRCA mutation and develop cancer in 5-10 years? At least provide the familiar history of all cases + mention this lack of matching in the limits of your study.

Best of luck with your future research!
